

# Grazing elevates litter decomposition but slows nitrogen release in an alpine meadow

Yi Sun[1,3], Xiong Zhao He[2], Fujiang Hou[1], Zhaofeng Wang[1], and Shenghua Chang[1]

[1] State Key Laboratory of Grassland Agro-ecosystems, College of Pastoral Agriculture Science and Technology,

Lanzhou University, Lanzhou, 730000, Gansu, China

[2] School of Agriculture and Environment, College of Science, Massey University, Private Bag 11-222, Palmerston

North, New Zealand

[3] State Key Laboratory of Cryospheric Sciences, Cold and Arid Regions Environmental and Engineering

Research Institute, Chinese Academy of Sciences, 320 Donggang West Road, Lanzhou 730000, China

*Correspondence to*: Fujiang Hou (cyhoufj@lzu.edu.cn)

**Abstract.** Nutrient cycling is a key ecosystem function whereby the processes of litter decomposition and N

release in the soil-plant interface are vitally important but still not clear in the alpine ecosystems. We carried out

a 3-year study to improve our understandings in nutrient cycle and develop strategies for restoring the degraded

grasslands on the Qinghai-Tibetan Plateau. We established the grazing (GP) and grazing exclusion paddocks

(GEP), then identified litter species composition and analyzed litter and soil chemical characteristics. Litter

decomposition and N release were monitored by incubating litter 'in situ' and across paddocks over 799 days. We

found that grazing did not change plant species composition, but increased litter N; while grazing exclusion

increased litter mass of palatable species and promoted soil organic carbon. Litter decomposed faster in GP, while

N release was faster in GEP. Incubation site environment had more but litter source had less impact on litter

decomposition and N release. Therefore, grazing and grazing exclusion had different impacts on litter



decomposition and N release but both elevated nutrient cycle. The implications of our findings in restoring the
degraded grasslands on the Qinghai-Tibetan Plateau were discussed.
**1 Introduction**
Qinghai-Tibetan Plateau represents an important eco-region in China (Wen et al., 2010), where the alpine
grasslands cover more than 85% of total area and are regarded as the major natural pastures (Dong et al., 2010).
However, the ecosystems in this region have continuously suffered from severe degradation mainly driven by
climate change, overgrazing, over-cultivation and poor management (Han et al., 2008; Li et al., 2009; Wu et al.,
2009; Feng et al., 2010; Wu et al., 2010), with an increasing degradation rate of 1.2-7.44% per year (Ma et al.,
2007). Since 1990's, the restoration of degraded grasslands has attracted highly attention (Kang et al., 2007; Han
et al., 2008), and some efforts have recently focused on the grassland restoration and maintenance by increasing
aboveground plant abundance (Niu et al., 2009) and biodiversity (Wu et al., 2009; Niu et al., 2010), and improving
soil property (Cao et al., 2004; Wu et al., 2010; Sun et al., 2011). However, litter decomposition and N release,
the key factors regulating the nutrient cycle and availability in the soil-plant interface (Carrera and Bertiller, 2013),
are still not clear in the alpine ecosystems (Luo et al., 2010; Zhu et al., 2016). We carried out this study to
investigate the effect of grazing on plant community, litter quality, litter decomposition and N release in an alpine
meadow, aiming to improve our understandings in the nutrient cycle and to develop management strategies for
restoration of ecological function in alpine pastures.

It is generally known that herbivore grazing may induce short-term ecophysiological changes in overall

litter quality as well as longer-term shifts in plant community composition. At the short-term ecophysiological
level, herbivore grazing may promote the plant species producing high-quality litter (Holland and Detling, 1990;
Sirotnak and Huntly, 2000; Olofsson and Oksanen, 2002; Semmartin et al., 2004, 2008). Because the loss of plant



tissues caused by grazing may favour the grazed species with a higher re-growth rate and greater nutrient contents
in plant tissues due to the higher nutrient uptake (see Holland and Detling, 1990; Olofsson and Oksanen, 2002;
Semmartin et al., 2008). At the long-term community level, selective foliar grazing of herbivores may alter the
competitive interaction and recruitment patterns of plant species, which will change their abundance and life form
structure (Bardgett and Wardle, 2003; Semmartin et al., 2008; Wu et al., 2009; Niu et al., 2010). For instance,
herbivores usually concentrate on the most palatable plants (e.g., species with high nutrient and low fibre contents),
which will favour dominance by the less unpalatable species (Garibaldi et al., 2007), resulting in the high inputs
of low-quality litter to soil and thus a reduction of decomposition rate, nutrient availability and nutrient cycling
(Ritchie and Knops, 1998; Moretto et al., 2001; Olofsson and Oksanen, 2002). However, empirical evidences of
variances in litter quality input and soil nutrient cycle caused by grazing are still scarce and controversial
(Garibaldi et al., 2007).
It is supposed that the higher nutrient content in plant tissue usually results in the faster litter decomposition,
and the high nutrient mineralization and availability in soil (Olofsson and Oksanen, 2002). At the ecosystem scale,
the chemical characteristics of plant litter, for example the carbon:nitrogen ratios (C:N) and/or nitrogen and lignin
content, are often regarded as the indicators of litter quality (Aerts, 1997; Strickland et al., 2009). Many studies
have demonstrated a positive correlation between litter decomposition rate and N content, or a negative
relationship between litter decomposition rate and initial lignin content and C:N or lignin:N ratios ratios (e.g.,
Wardle et al., 2002; Aerts et al., 2003; Semmartin et al., 2004; Garibaldi et al., 2007; Luo et al., 2010; Vaieretti et
al., 2013).
Except litter quality (its chemical composition), the climate (mainly temperature and humidity) and
decomposing organisms (their abundance and activity) are the two main factors controlling litter decomposition
(Coûteaux et al., 1995; Aerts, 1997; Semmartin et al., 2004; Keeler et al., 2009; Berg and McClaugherty, 2014;



Zhu et al., 2016). Generally, climate is the dominant factor over litter quality in areas subjected to unfavourable
weather conditions (Coûteaux et al., 1995), because decomposer activity regulating litter decomposition is largely
temperature and moisture dependent (De Santo et al., 1993). Usually the activity of decomposers increases with
temperature (Coûteaux et al., 1995); and as soil moisture level rises, the metabolic activity of decomposers
normally increases until an optimum plateau is reached (Orsborne and Macauley, 1988). Thus it is likely that the
optimum response of decomposers to temperature and moisture is determined largely by the local climate (De
Santo et al., 1993). However, under the favourable conditions, litter quality may largely prevail as the regulator
and remain important until the late decomposition stages (Coûteaux et al., 1995).
Most studies on litter decomposition affected by the herbivores usually focus on the forest, grassland or
crop ecosystems in the temperate areas, largely ignoring that in the alpine zones. Moreover, to date our knowledge
on the decomposition of litter particular that of mixed-litter in the grasslands in the alpine regions is still lacking,
though some efforts have been made (e.g., Wu et al., 2009; Luo et al., 2010; Zhu et al., 2016). In this study, we
firstly set up the grazing (GP) and grazing exclusion paddocks (GEP) and examined the subsequent plant
composition, litter quality and soil characteristics. We then subsequently investigated whether litter quality had
any effects on litter decomposition, N release and soil property by collecting litter mixtures from both GP and
GEP and incubating them 'in situ' and across GP and GEP. Based on the knowledge outlined above, we tested:
(1) whether grazing may improve litter quality with higher nutrient content (i.e., N) and change plant community
with lower biomass of palatable species detected whereas the opposite is the case in GEP, and (2) whether grazing
will elevate litter decomposition and N release and thus improve soil properties whereas the opposite is the case
in GEP. Results of the present study may provide insight into our understandings in nutrient cycle in the alpine
regions in general, which may also further provide a scheme to assist in the development of strategies for restoring
the degraded grasslands on the Qinghai-Tibetan Plateau in particular.




## 2 Materials and methods

### 2.1 Experimental site

This study was conducted in a typical alpine meadow, eastern Qinghai-Tibetan Plateau, NW China (N33°59′,
E102°00′, altitude 3,500 m above sea level). The mean annual temperature is 1.2°C, ranging from -10°C in January
to 11.7°C in July, with approximately 270 days of frost per year. The mean annual precipitation is 620 mm over
the last 35 years, occurring mainly during the short and cool summer (Niu et al., 2010). The years during this
study (i.e., 2009 ~ 2012) were climatically typical (Sun et al., 2015), and the mean temperature and total
precipitation of each month during the experiments were showed in Supplemental Fig. 1.
The pasture selected for experiments is larger than nine hectares (including experimental and buffer areas)
and regularly used for Tibetan sheep and yak grazing during the grazing seasons (May ~ October). The slopes are
less than 5%, which is the gentle topography of the area, and the soil attributes in the experimental areas were
similar after a long-term grazing history with the same grazing pattern. The soil type in the experimental area is
alpine meadow soil, similar to the primarily Mat-Cryic Cambisols (Wu et al., 2010).

### 2.2 Litter composition and quality

To measure the annual litter composition and determine whether plants could recover without grazing, three
grazing (GP, 30 m × 20 m) and three grazing exclusion paddocks (GEP, 100 m × 200 m) were established when
all aboveground plants are dormant in October 2009. Grazing in GP started with an optimal moderate stocking
rate of 4 Tibetan sheep/ha since April 2010. The mean body weight of sheep was about 38 kg when used for the
experiment. In October 2010, 20 quadrats (0.5 m × 0.5 m) were randomly established within the GP or GEP, and
three sampling methods [i.e., half alongside, half along diagonal, and two sub-quarters (0.25 m × 0.25 m) along



diagonal; see Supplemental Fig. S2] were designed to minimize the sample variance caused by the uneven litter
distribution and to ensure the similar composition and quality of litter used for this and next experiments. The
aboveground portion of all dormant plants (i.e., litter) from each quadrat was sampled for two purposes: (1)
measurement of litter composition and quality, and (2) measurement of litter decomposition and N release.
To measure the litter composition, litter of different species was identified and then separated into two
groups according to the palatability by the Tibetan sheep (Niu et al., 2009, 2010; Wu et al., 2009; see
Supplementary Table 1): (1) palatable species - preferred and desirable species, and (2) unpalatable species -
undesirable and toxic species.
To measure the litter quality, two treatments were tested: (1) GP-litter – litter of all species from
aboveground was collected from the GP, (2) GEP-litter – litter of all species from aboveground was collected
from the GEP. Litter from a quadrat was mixed after the measurement of litter composition and then oven-dried
at 60°C for 48 h. The dry litter was ground and stored in a zip-lock bag with 10 g per bag. There were six replicates
for each treatment. The contents of lignin, cellulose and hemicellulose were measured by following van Soest et
al. (1991). Organic carbon concentration (C) was measured by the $K_2Cr_2O_7$-$H_2SO_4$ oxidation method of Walkley-
Black (Nelson and Sommers, 1996). Briefly, a known weight of plant sample was treated with potassium
dichromate in the presence of concentrated sulfuric acid and digested at a low temperature ($\approx 30°C$) for one hour.
The excess of potassium dichromate did not reduce the organic matter, as it was titrated back against a standard
solution of ferrous sulfate. The C was then calculated on the basis of the quantity of ferrous sulfate consumed
(Chen et al., 2016). The total Kjeldahl nitrogen (N) and total phosphorus (P) were analyzed using a FIAstar 5000
flow injection analyzer (Foss Tecator, Högnäs, Sweden) (Chen et al., 2016). We also calculated the ratios of C:N,
lignin:N, cellulose:N and hemicelluloses:N.
The soil environment modified by land use pattern is thought to be another important driver of variations



of litter composition and quality, and thus litter decomposition and N release. Therefore, we also examined the
effects of grazing and grazing exclusion on soil characteristics. We randomly selected five sampling sites from
each experimental paddock, and soil samples (n = 30 in total) were collected at depth 0-10 cm using a bucket
auger (10 cm in diameter) in October of 2010, 2011 and 2012. The same methods used to test litter quality (i.e.,
Nelson and Sommers, 1996; Chen et al., 2016) were applied to estimate the soil organic carbon (SOC), total
nitrogen (TN) and total phosphorus (TP).

**2.3 Litter decomposition and N release**
In this experiment, we carried out four treatments: (1) GP-GP, litter of all species was collected from and incubated
in the GP; (2) GEP-GEP, litter of all species was collected from and incubated in the GEP; (3) GP-GEP, litter of
all species was collected from the GP but incubated in the GEP; (4) GEP-GP, litter of all species were collected
from the GEP but incubated in the GP. The treatments 1 and 2 were referred as "in situ" incubation treatments.
The treatments 3 and 4 were across incubation treatments which were to improve our better understanding "home-
field advantage" effect on litter deposition (John et al., 2011).

For each sample soil particles attached to litter were cleaned off with a soft brush, and samples were air-dried

for three days. Dry litter collected from each quadrat were cut to ≈ 5 cm length and 10 g litter was packed into a
nylon litter-bag (15 cm × 20 cm with mesh size of 0.35 mm) (Cornelissen, 1996), which may prevent any loss of
material and has no effect on litter decomposition (Cornelissen et al., 1999). On 20[th] Oct 2010, the packed litter
was incubated above the soil surface by fastening to the ground surface with four steel stakes to avoid being
removed by the sheep and small animals (Vaieretti et al., 2013). The small animals were the plateau pika,
*Ochotona curzoniae* (Hodgson) in the present study. For each treatment, 24 packed litter-bags were incubated
with 20 cm apart from each other to reduce the mutual interference. Three litter bags were retrieved after a



incubation period of 56, 141, 247, 391, 444, 582, 695 or 799 days (i.e., on 15th December 2010, 10th March, 24th
June and 4th September 2011, and 7st January, 24th May, 14th September and 27th December 2012, respectively).
There were a total of 144 packed litter-bags used in this experiment. Retrieved litter was brought back to the
laboratory and cleaned by removing any extraneous material attached. They were weighed after being oven-dried
at 60ºC for 48 h. Samples were ground and stored in a zip-lock bag for further chemical analyses as mentioned
above. We estimated the litter decomposition and N release as the percentage of dry weight remaining at the end
of each incubation period (Cornelissen et al., 1999; Vaieretti et al., 2013).

**2.4 Statistical analyses**
A goodness-of-fit test (Shapiro-Wilk test, Univariate Procedure) was used to test the normality of data before
mean comparison using analysis of variance (ANOVA, GLM Procedure). All those data were normally distributed.
Data on litter quality (Table 1) were analyzed using ANOVA followed by Tukey's Studentized multiple range
test. While the difference between the palatable and unpalatable litter quantity in GP or GEP was compared by
paired-t test (TTEST Procedure) (Fig. 2). Data on the proportion of litter mass or N remaining (Fig. 3-5), litter
quality (content of organic carbon, nitrogen, phosphorous and so on) (Table 1), and soil SOC, TN and TP (Fig. 1)
were analyzed using ANOVA followed by least significant difference test (LSD test) for multiple comparisons.
The decay rate ($k$) of litter mass during the incubation period (Table 2) was assessed using a negative
exponential model according to Swift et al. (1979): $y = a*e^{-(t*k)}$, where $y$ is the dry mass of litter remaining in the
litter bags at time $t$ (days), $a$ is the initial litter mass. The difference in decay rate between treatments was compared
according to Julious (2004), i.e. there is no significant difference in decay rate if their 83.4% CL overlap. The
decay rate and 83.4 % CI were estimated by fitting the negative exponential model to a nonlinear least square
regression model (NLIN Procedure).





To provide quantified information of how environmental conditions (i.e., decomposition sites, the GP and

GEP) and litter source (i.e., litter collected from the GP and GEP) affecting the final litter decomposition and N

release, a multivariate regression model (GLM Procedure) employed by Vaieretti et al. (2013) was used (Table

3): litter decomposition or N release = site + litter source + site × litter source + $\epsilon$, where $\epsilon$ is the model error. The

proportion of site environment, litter source and their interaction contributing to variability of litter decomposition

or N release was calculated as: their sum of square divided by the total sum of square. The Type I sum of square

was used because of the balanced design of this experiment. All analyses were done using SAS 9.3 (SAS Institute

Inc., Cary, NC, USA). Rejection level of $H_0$ was set at $\alpha < 0.05$. Mean (± SE) was presented in Fig. 1-5.

**3 Results**

**3.1 Litter composition and quality and soil property**

Fifty-five plant species (mostly forbs and graminoid grasses with several legumes and sedges) were identified,

and they presented in both GP and GEP, except the *Gentiana macrophylla* Pallas, which was only found in the

GP (Supplemental Table 1). However, even though the annual litter mass of unpalatable species in both GP and

GEP was similar (ANOVA: $F_{1,38} = 3.43$, $P = 0.0717$), litter mass of palatable species was significantly greater in

the GEP than in the GP (ANOVA: $F_{1,38} = 75.32$, $P < 0.0001$), which contributed significantly more to the total

litter mass in the GEP than in the GP (ANOVA: $F_{1,38} = 114.66$, $P < 0.0001$) (Fig. 1). The litter mass was not

significantly different between palatable and unpalatable species in the GP (Paired-t test: $t_{19} = 0.96$, $P = 0.3510$);

however, in the GEP, litter mass of palatable species was significantly greater than that of unpalatable species

(Paired-t test: $t_{19} = 7.17$, $P < 0.0001$) (Fig. 1).

As shown in Table 1, litter collected from GP had significantly higher C and N but significantly lower

hemicelluloses and hemicelluloses:N than that collected from GEP. No significant difference was found in other





compounds or compound ratios between different litter sources though the cellulose, C:N, lignin:N and
cellulose:N were lower in litter collected from GP (Table 1).
The concentrations of soil TN and TP were not significantly different between GP and GE for each year
(LSD = 0.0002~0.0015 and 0.0003~0.0004 for TN and TP, respectively; $P > 0.05$) (Fig. 2a-b). Similarly, there
was no significantly difference in SOC concentration between the GP and GE in 2010 and 2011 (LSD = 0.0169
and 0.0111 for 2010 and 2011, respectively; $P > 0.05$), while in 2012 SOC was significantly higher in the GE than
in the GP (LSD = 0.0138, $P = 0.0279$) (Fig. 2c).

**3.2 Litter decomposition**
The proportion of litter mass remaining significantly decreased with incubation duration (LSD test: LSD = 3.23
~ 4.96, $P < 0.0001$) (Fig. 3). It is found that the lower environmental temperature ($< 0°C$) between November and
March (Supplemental Fig. 1) might have significantly slowed the litter decomposition (Fig. 3); but increasing
temperature since April (Supplemental Fig. 1) significantly accelerated litter decomposition (Fig. 3). The time
period required to achieve 50 % decomposition of litter mass was about 19 months in GP-litter, which was faster
than that in GP-GEP-litter (i.e., about 23 months) (Fig. 3).
As shown in Table 2, the decomposition rate ($k$) of litter incubated in the GP was significantly higher than
that in the GEP (non-overlap of 83.4% CL), i.e., for the 'in situ' treatments $k$ in GP-GP > $k$ in GEP-GEP) and for
the across treatments $k$ in GEP-GP > $k$ GP-GEP. The final proportion of litter mass remaining was significantly
lower in GP-GP and GEP-GP than in GP-GEP (LSD test: LSD = 2.51, $P < 0.0001$) (Fig. 5a).

**3.3 N release**
Compared to litter decomposition, the dynamics of N release were more complicate. Generally, the percentage of





total N release did not significantly change during the first winter when temperature < 0°C, except that it
significantly increased during the incubation period of December 2010 and March 2011 (first winter) in treatments
of GEP-GP (Fig. 4). Since January in the second winter (2012), the percentage total N remaining significantly
decreased until the end of experiments (Fig. 4). The final proportion of total N remaining was significantly higher
in the GEP-GP and significantly lower in the GEP-GEP and GP-GEP (LSD test: LSD = 5.36, P = 0.0002) (Fig.
5b).

**3.4 Contribution of site environment and litter source to litter decomposition and N release**
The multivariate regression model indicates that both site environment and litter source significantly affected litter
decomposition and N release with various contributions (Table 3). Site environment contributed respectively near
25% and 50% more to litter decomposition and N release than did litter source (Table 3). Furthermore, the model
predicts that GP resulted in significantly greater litter decomposition (8.13%) but significantly fewer N release
(9.73%) than did GEP ($F_{1,8}$ = 62.48 and 57.49 for litter decomposition and N release, respectively; P < 0.0001).
Results show that litter collected from GEP decomposed significantly faster (2.5%) but released N significantly
slower (9.27%) than that collected from GP ($F_{1,8}$ = 34.99 and 15.80 for litter decomposition and N release,
respectively; P < 0.01).

Significant interaction of site environment and litter source was found on litter decomposition ($F_{2,8}$ = 7.10,

P = 0.0286), i.e., litter collected from GEP but incubated in GP decomposed significantly faster (also see Fig. 5a).
However, interaction of site environment and litter source was not significant on N release ($F_{2,8}$ = 2.76, P = 0.1350).

**4 Discussion**
**4.1 Litter Composition and Quality**



Grazing of herbivores may indirectly alter the species composition and functioning of grasslands by inducing
shifts in plant competitive interactions and recruitment patterns, and thus changes in species abundances and life
form structure (Bardgett and Wardle, 2003; Garibaldi et al., 2007; Semmartin et al., 2008; Wu et al., 2009; Niu et
al., 2010; Chaneton, 2011). However, our results indicate that herbivore grazing did not alter plant community
composition in terms of species inventory, as species found in the GEP mostly presented in the GP. According to
Sun et al. (2011), herbivores caused changes in species composition is grazing-intensity dependent. Thus our
results imply that a moderate stocking rate of 4 Tibetan sheep/ha with a short annual grazing period (i.e. from
April to October 2010) in the GP did not change the species composition. To confirm this conclusion, a long-term
grazing exclusion experiment will provide more confident proofs.

However, our results show that herbivore grazing significantly altered species composition in terms of

species abundance or palatability with significantly less palatable litter produced in the GP than in GEP (Fig. 1).
The low mass of palatable species may attribute to two causations. First, on the Qinghai-Tibetan Plateau herbivore
grazing will can in the short plant height and small leaf area of palatable species (Sun et al., 2011). This may be
an effective ecological strategy of palatable species to avoid being grazed and the continuous disturbance, through
increasing resistance to grazing with lower competition for light (van der Wal et al., 2000; Falster and Westoby,
2003; Niu et al., 2009, 2010). Second, most palatable species on the Qinghai-Tibetan Plateau (mostly the grasses
and sedges, see Supplemental Table 1) are tall and more likely grazed by preferential grazing (Sun et al., 2011),
subsequently resulting in the lower mass of palatable species in the GP. Furthermore, our results show that there
was no significant difference in the litter mass of unpalatable species between the GP and GEP (Fig. 1),
disagreeing with the assumption that removing the canopy of palatable species will allow the intra- and inter-
specific competition for light which ultimately favours the establishment of short, less-palatable species
(Sternberg et al., 2000; Pavlů et al., 2008; Wu et al., 2009; Sun et al., 2011).




It is well known that litter quality is usually determined by the content of different chemical compounds
such as soluble C, N and P, as well as lignin or lignin:N ratio, and litter of high quality usually has higher N
content but lower lignin and lignin:N ratio (e.g., Aerts, 1997; Olofsson and Oksanen, 2002; Wardle et al., 2002;
Garibaldi et al., 2007; Strickland et al., 2009). In this study, grazing might have improved litter quality at least to
some degrees by significantly increasing the N content and potentially lowering the hemicellulose content and
C:N, lignin:N, cellulose:N and hemicellulose:N ratios (Table 1). Therefore, our results agree with previous studies
that grazing may promote litter quality (e.g., Sirotnak and Huntly, 2000; Olofsson and Oksanen, 2002; Semmartin
et al., 2004).

**4.2 Litter decomposition**
For a given climate region, the ecological processes of litter decomposition are regulated by incubation
environment (i.e., grazing/grazing exclusion and soil property in this study) and litter quality. Our results indicate
that herbivore grazing played a major role in litter decomposition on the Qinghai-Tibetan Plateau. Firstly many
studies have demonstrated that litter quality is one of the most important factors affecting the litter decomposition,
and litter with higher N content but lower lignin and lignin:N ratio will decompose faster (Aerts, 1997; Olofsson
and Oksanen, 2002; Wardle et al., 2002; Garibaldi et al., 2007; Strickland et al., 2009). By following this line, it
may be assumed that regardless of incubation site, litter collected from the GP will decompose faster than that
collected from GEP. It is true for the "in situ" incubation treatments, i.e., decomposition rate was significantly
greater in GP-GP than in GEP-GEP; however, for the across incubation treatments opposite results were found,
i.e., it was significantly greater in GEP-GP than in GP-GEP (Table 2 and Fig. 5a). In fact the multivariate
regression model shows that litter collected from the GEP had less mass remaining (i.e., 2.5%) compared to that
collected from the GP. In the study of tissue and fertilizer N affecting decomposition of conifer litter, Perakis et



al. (2012) revealed that high initial litter N slows decomposition rate in both early and late stages in unfertilized
plots. Fertilizer N is usually not applied in the pastures on Qinghai-Tibetan Plateau (F.H personal comment).
Moreover, Aerts (1997) suggests that litter chemistry is not a good predictor on litter decomposability in the cold
temperate region. Our experimental area locates 3,500 m above sea level with a typical alpine climate having a
mean annual temperature of 1.2°C and ranging from -10°C in January to 11.7°C in July.

Alternatively, for a given litter source (i.e., collected from GP or GEP) a greater decomposition rate might

be caused by the grazing herbivores when litter was incubated in GP (Table 2 and Fig. 5a), because grazing
herbivores may modify site conditions for litter turnover through physical changes in the soil through herbivore
activities, such as trampling and urine/dung deposition (Takar et al., 1990; Fahnestock and Knapp, 1994; Luo et
al., 2010). Such argument is demonstrated by the results of multivariate regression model: (1) the significantly
greater effect of incubation site environment (≈ 25%) than that of litter source or quality, and (2) the the
significantly greater effect of GP (8.13%) than that of GEP (Table 3). Grazing elevates litter decomposition is
widely reported (e.g., Takar et al., 1990; Fahnestock and Knapp, 1994; Garibaldi et al., 2007; Semmartin et al.,
2008; Luo et al., 2010).

On the Qinghai-Tibetan Plateau, Wu et al. (2009) report that a long-term (9 years) grazing exclusion favors

the increase of soil total nitrogen, soil organic matter, soil organic carbon, soil microbial biomass carbon and soil
carbon storage. It is intersting that in the present study, only the SOC significantly increased after three-years
grazing exclusion (Fig. 2c); however, grazing exclusion did not significantly modify soil proporty in terms of TN
and TP. The increase of SOC in GEP may be because grazing exclusion prevents the reduction of outflow of
palatable litter by the herbivores (Fig. 1), and the organic C locked within plant tissues will be returned to the soil
during litter decomposition (Bardgett and Wardle, 2003; Wu et al., 2009). Holland and Detling (1990) and Ågren
et al. (1999) state that increasing carbon availability in soil will promote decomposer growth and activity even at





the low nitrogen concentrations. However, the expected results, i.e. significant higher litter decomposition rate
caused by the possible increasing decomposer mass and/or activity in the GEP (Wu et al., 2009), were not observed
(Table 2 and Fig. 5a). Therefore, during a relatively short period of time the soil property is unlikely significantly
changed through herbivore urine/dung deposition and may have less effect on litter decomposition under cool
environments on the Qinghai-Tibetan Plateau.

Our study site is a typical alpine meadow with a long and cold winter and spring (Supplemental Fig. 1).

Climate is the dominant factor regulating litter decomposition and nutrient cycling in such area, and the activity
of decomposers may be inhibited during the cold seasons (Coûteaux et al., 1995). These may have interpreted the
slow progresses of litter decomposition (Fig. 3) during the first winter and spring seasons when the low
temperature inhibited decomposer growth and activity. While the optimum response of decomposers to
temperature and moisture occurs during summer and early autumn, elevating the litter decomposition rate (Fig. 3)
when both temperature and precipitation simultaneously increase (Supplemental Fig. 1), as reported by Orsborne
and Macauley (1988) and De Santo et al. (1993). Because of the slow biological processes, the decomposition
rates estimated in this study ($k = 0.38 \sim 0.49$, Table 2) is lower than global ones ($k \approx 0.75$) with similar latitudes
(N30 ~ 40°) (Zhang et al., 2008).

**4.3 N release**
N release is more complex compared to litter decomposition. N release from litter may involve any one or both
procedures of N immobilization and N mineralization, where the former results in the accumulation of N in the
litter and the latter causes the release of N from the litter (Manzoni et al., 2008). Swift et al. (1979) and Berg and
McClaugherty (2014) report that the biological decomposition of litter is mainly carried out by microbial
decomposers, including bacteria and fungi, which have higher N/C value compared with most litter types. This
property of decomposers creates a high N demand for growth. Therefore, Manzoni et al. (2008) state that even



though a considerable fraction of assimilated C is respired, the decomposers often still require some inorganic N
uptake during at least the early phases of decomposition. The increase of N remaining in the GP-GP, GEP-GEP
and GEP-GP at beginning of incubation season (i.e., from October to December 2010) in the present study (Fig.
4a-c) provided direct evidence supporting the above assumption of Manzoni et al. (2008).

However, our results also show an increase of N remaining from a lower content in June 2011 to a higher

content in September 2011 (Fig. 4). It may be because that the fast decomposition rate of litter during summer
(Fig. 3) results in utilization of C by decomposers which increases N:C ratio. Similarly, Bosatta and Balesdent
(1996) and Manzoni et al. (2008) have demonstrated a positive correlation between the decomposer N:C and
respiration rate (the efficient carbon use) defining the actual nutrient requirement of decomposers; thus the higher
respiration rate of decomposers during summer will elevate N:C ratio. Manzoni et al. (2008) also indicate that
when N:C is high, large amounts of mineral N are immobilized, which will in turn increase the litter N
concentration. However, N release starts after the accumulative N level reaches a critical level (Berg and Staaf,
1981), causing a significant decrease of N remaining during April ~ June 2011 and during the end incubation
period (May ~ December 2012) (Fig. 4).

Based on the above knowledge, inverse patterns of litter decomposition and N release were found, i.e., the

greater litter decomposition rate was (Fig. 5a), the lower N release rate became (Fig. 5b). Many studies of litter
decomposition have found an increase in relative nitrogen concentration in litter as the litter decomposition
proceeds (e.g., Aber and Melillo, 1980; Fahey et al., 1991; Gallardo and Merino, 1992). Aber and Melillo (1980)
state that an inverse-linear relationship between the remaining biomass and nitrogen concentration in litter is
validated for a large number of litter decomposition studies.

The inverse patterns of N release compared to litter decomposition is also demonstrated by the results of

multivariate regression model: (1) although the incubation site environment had significantly greater effect (≈



50%) on N release than did litter source or quality (Table 3), N release was significantly lower in GP (9.27%)
than in GEP; (2) litter collected from GP had significantly lower N remaining than did GEP. The latter result
contradicts the assumption of Berg and Staaf (1981) that the relatively higher initial nitrogen concentration (see
Table 1 for GP litter) may contribute to the higher nitrogen immobilization in litter. The cause is unknown but
this phenomenon indicates that litter chemistry may not be a good predictor on litter N release as Aerts (1997)
suggests for litter decomposability in the cold temperate region.

**5 Conclusion**

Our findings provide insight into our understandings in the litter decomposition and N release affected by

herbivore grazing in a typical alpine meadow. Compared to grazing exclusion, grazing with a moderate stocking
rate could improve litter quality with higher N but lower hemicellulose and hemicellulose:N ratio, and herbivore
activity significantly elevated litter decomposition rate regardless of litter source. While grazing exclusion was
more likely to maintain significantly more palatable litter, promoted N release and increased SOC. The different
effects of livestock grazing and grazing exclusion on litter decomposition and N release may have implications in
the management of alpine meadows on the Qinghai-Tibetan Plateau. For example, grazing exclusion for one
season following by livestock grazing will be a good option which allows plant recovery, promotes N release,
elevates litter decomposition, and thus contributes to the restoration of degraded grasslands.

*Author contributions.* YS and FH designed the experiments, YS, ZW and SH Chang performed research and
collected data. XZH and YS analysed data and prepared the manuscript, and all authors contributed to the writing.

*Competing interests.* The authors declare that they have no conflict of interest.



*Acknowledgments.* This study was financially supported by the National Key Project of Scientific and Technical
Supporting Programs (2014CB138706), National Natural Science Foundation of China (No. 31672472), Program
for Changjiang Scholars and Innovative Research Team in University (IRT13019) and the independent grants
from the State Key Laboratory of Grassland Agro-ecosystems (SKLGAE201708).

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





**Table titles**
**Table 1.** Mean (± SE) initial contents of carbon (C, mg g$^{-1}$), nitrogen (N, mg g$^{-1}$), phosphorus (P, mg g$^{-1}$), lignin (mg g$^{-1}$),
cellulose (mg g$^{-1}$) and hemicellulose, and C:N, lignin:N, cellulose:N and hemicellulose:N ratios in litter*.

**Table 2.** Estimated litter decay rate ($k$) in different incubation environments.

**Table 3.** Estimated contribution (%) of incubation site environment (Site, GP and GEP) and litter source (Source, GP and
GEP) to litter decomposition and N release by multivariable regression model.*

















**Figure captions**
**Figure 1.** Estimated mean (± SE) annual litter mass in GP and GEP. For each category, columns with different letters are
significantly different (ANOVA: P < 0.05).

**Figure 2.** Comparison of soil property between GP and GEP: (**a**) TN, (**b**) TP and (**c**) SOC. *Significant difference was only
found between GP and GEP for SOC in 2012 (P < 0.05).

**Figure 3.** Decrease of litter mass remaining with incubation time. For each treatment columns with the different letters are
significantly different (P < 0.05). Vertical bar is the LSD value. Grey lines under months indicates the air temperature < 0 ℃.

**Figure 4.** Dynamics of total N remaining with incubation time. For each treatment columns with the different letters are
significantly different (P < 0.05). Vertical bar is the LSD value. Grey lines under months indicates the air temperature < 0 ℃.

**Figure 5.** Percentage of litter mass (**a**) and total N remaining (**b**) at the end of experiments. Columns with the different letters
are significantly different (P < 0.05). Vertical bars are the LSD values.











**Table 1.** Mean (± SE) initial contents of carbon (C, mg g$^{-1}$), nitrogen (N, mg g$^{-1}$), phosphorus (P, mg g$^{-1}$), lignin
(mg g$^{-1}$), cellulose (mg g$^{-1}$) and hemicellulose, and C:N, lignin:N, cellulose:N and hemicellulose:N ratios in litter*.

| Compound | GP litter | | GEP litter | | LSD | P |
|---|---|---|---|---|---|---|
| C | 576.44 ± 4.20 | a | 553.03 ± 3.35 | b | 14.92 | 0.0121 |
| N | 7.41 ± 0.32 | a | 5.35 ± 0.67 | b | 2.05 | 0.0494 |
| P | 1.39 ± 0.27 | a | 1.05 ± 0.27 | a | 1.05 | 0.4197 |
| Lignin | 22.94 ± 4.57 | a | 18.83 ± 1.67 | a | 13.51 | 0.4456 |
| Cellulose | 328.61 ± 11.55 | a | 385.18 ± 19.27 | a | 62.38 | 0.0655 |
| Hemicellulose | 296.76 ± 6.82 | b | 324.56 ± 5.52 | a | 24.37 | 0.0340 |
| C:N | 78.08 ± 3.62 | a | 106.82 ± 13.96 | a | 40.05 | 0.1171 |
| Lignin:N | 3.15 ± 0.72 | a | 3.70 ± 0.74 | a | 2.87 | 0.6210 |
| Cellulose:N | 44.40 ± 1.35 | a | 75.23 ± 13.15 | a | 36.70 | 0.0800 |
| Hemicellulose:N | 40.12 ± 1.21 | b | 62.58 ± 7.69 | a | 21.61 | 0.0447 |

*Litter collected from different paddocks in GP or GEP was mixed well before test. Means with the different
letters in each row are significantly different (ANOVA: P < 0.05).



**Table 2.** Estimated litter decay rate ($k$) in different incubation environments.

| Treatment | $K \pm SE$ ($\times 10^{-3}$) | 83.4% CL ($\times 10^{-3}$) | $R^2$ | $F_{1,26}$ | P |
|-----------|------------------------|----------------------|-------|-------|-----|
| GP-GP | $1.34 \pm 0.04$ a | 1.28~1.40 | 0.9666 | 6716.09 | < 0.0001 |
| GEP-GEP | $1.20 \pm 0.04$ b | 1.14~1.27 | 0.9545 | 5646.09 | < 0.0001 |
| GEP-GP | $1.30 \pm 0.07$ ab | 1.12~1.44 | 0.9149 | 2382.19 | < 0.0001 |
| GP-GEP | $1.04 \pm 0.02$ c | 0.10~1.07 | 0.9809 | 8524.80 | < 0.0001 |

Mean ($\pm$ SE) followed by the different letters are significantly (non-overlap of 83.4% CL). The values of $R^2$, F
and P are estimated from the negative exponential model of Swift et al. (1979).



**Table 3.** Estimated contribution (%) of incubation site environment (Site, GP and GEP) and litter source (Source,
GP and GEP) to litter decomposition and N release by multivariable regression model.*

| Parameter | df | Type I SS | Contribution | F | P |
|---|---|---|---|---|---|
| *Litter decomposition* | | | | | |
| Site | 1 | 110.84 | 55.51 | 62.48 | < 0.0001 |
| Source | 1 | 62.06 | 31.08 | 34.99 | 0.0004 |
| Site×Source | 1 | 12.59 | 6.30 | 7.1 | 0.0286 |
| Error | 8 | 14.19 | 7.11 | | |
| *N release* | | | | | |
| Site | 1 | 466.25 | 68.39 | 57.49 | < 0.0001 |
| Source | 1 | 128.18 | 18.80 | 15.81 | 0.0041 |
| Site×Source | 1 | 22.41 | 3.29 | 2.76 | 0.1350 |
| Error | 8 | 64.88 | 9.52 | | |

* For litter decomposition: $F_{3,8}$ = 34.86, P < 0.0001, $R^2$ = 0.9289; for N release: $F_{3,8}$ = 25.35, P = 0.0002, $R^2$ =

603 0.9048.





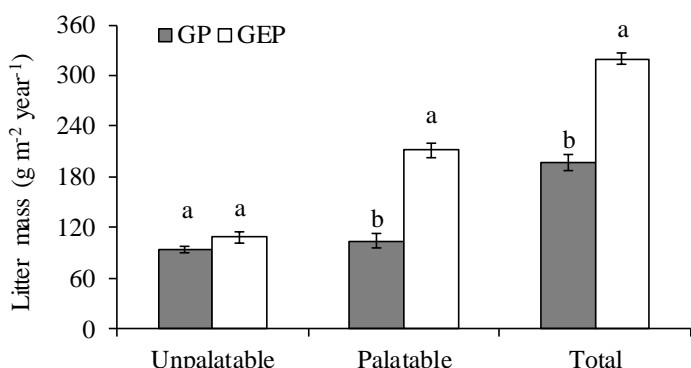

**Figure 1.** Estimated mean (± SE) annual litter mass in GP and GEP. For each category, columns with different
letters are significantly different (ANOVA: P < 0.05).



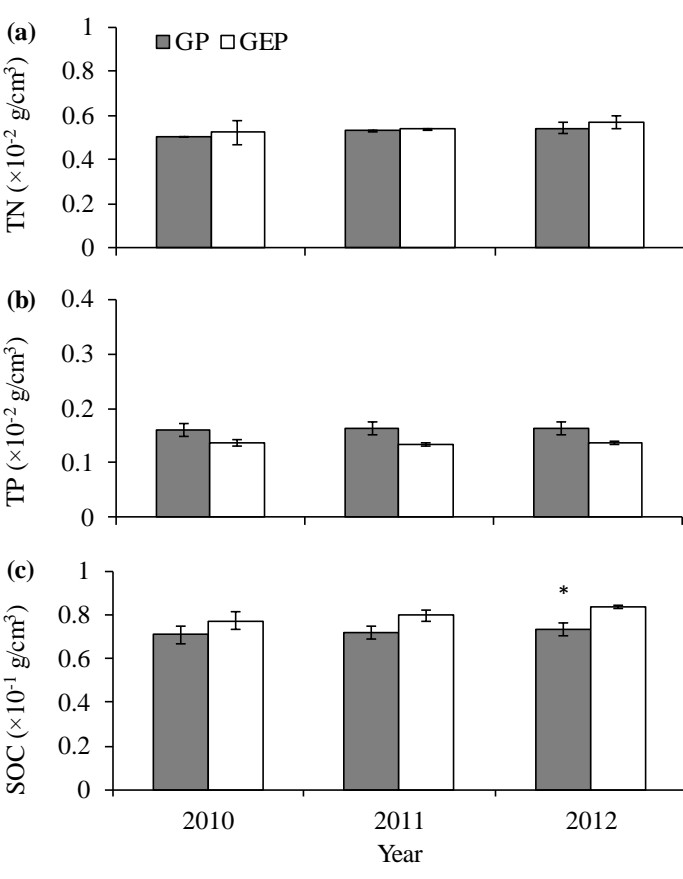


**Figure 2.** Comparison of soil property between GP and GEP: (**a**) TN, (**b**) TP and (**c**) SOC. *Significant difference

was only found between GP and GEP for SOC in 2012 ($P < 0.05$).



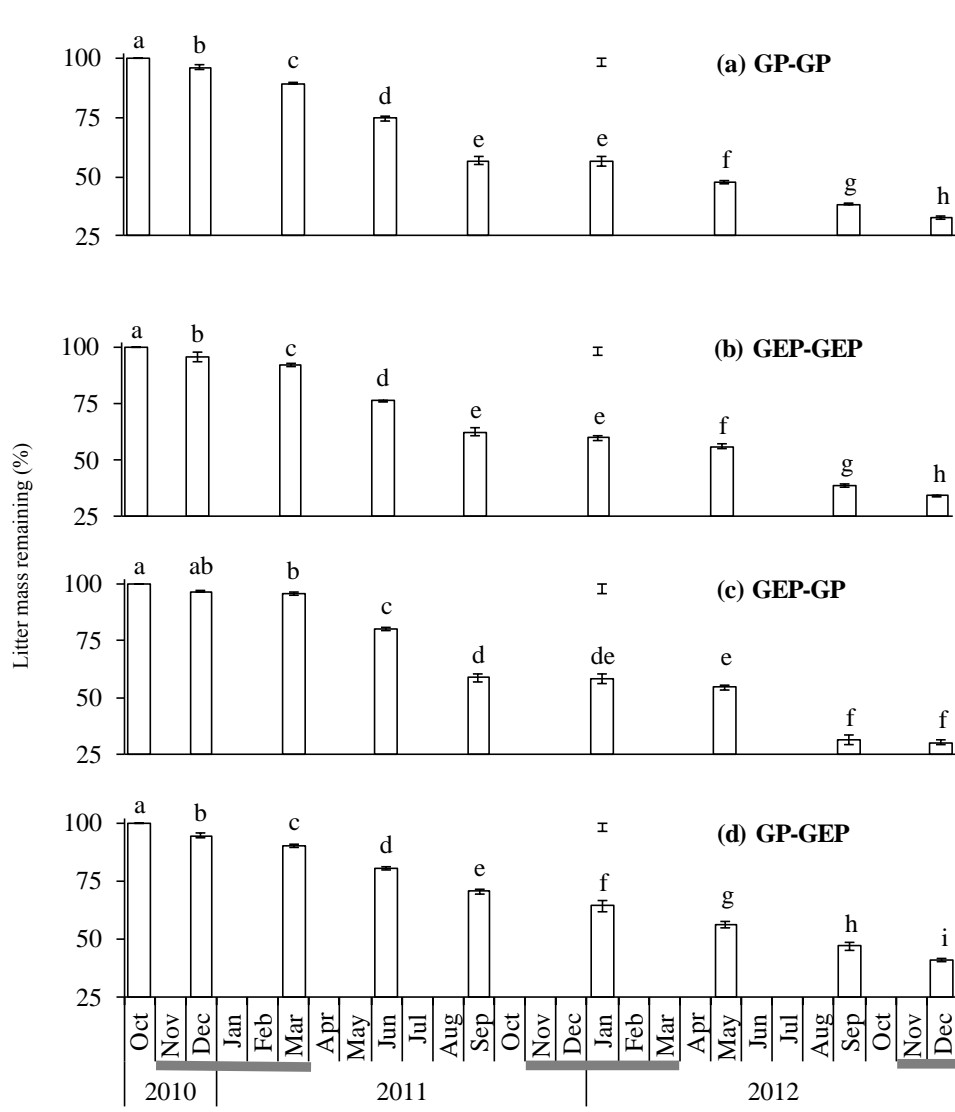

**Figure 3.** Decrease of litter mass remaining with incubation time. For each treatment columns with the different
letters are significantly different (P < 0.05). Vertical bar is the LSD value. Grey lines under months indicates the
air temperature < 0 ºC.



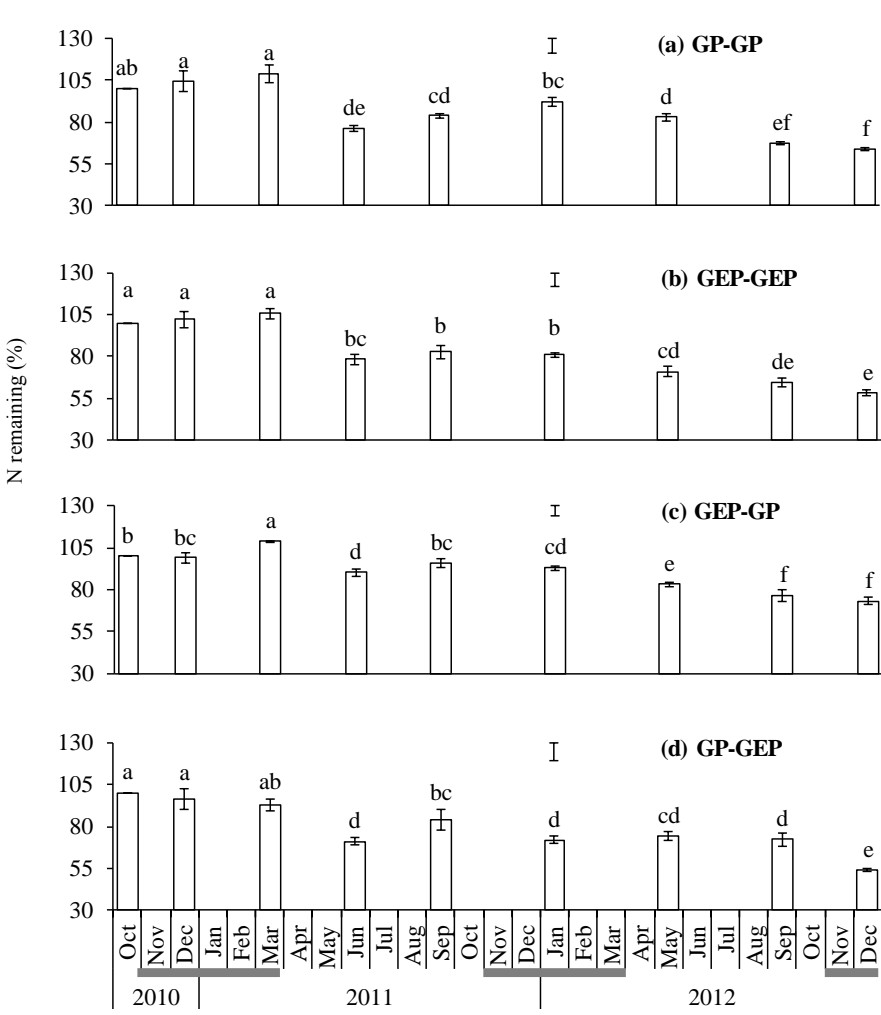

**Figure 4.** Dynamics of total N remaining with incubation time. For each treatment columns with the different letters are significantly different (P < 0.05). Vertical bar is the LSD value. Grey lines under months indicates the air temperature < 0 ºC.





**Figure 5.** Percentage of litter mass (**a**) and total N remaining (**b**) at the end of experiments. Columns with the
different letters are significantly different (P < 0.05). Vertical bars are the LSD values.