# Peer review of "Grazing elevates litter decomposition but slows nitrogen release in an alpine meadow"

_Biogeosciences, 2018_

## Referee Comment (RC1) · Anonymous Referee #1 · 28 Apr 2018

Comments for the authors

This study aims to test the effect of grazing on litter decomposition and N release in an alpine Tibetan grassland system. The focus of the manuscript is interesting, and can contribute to the management practices at local and regional scale. However, the authors did not show the novelty found with this work. I think that the authors need to express explicitly in the manuscript the importance of the results found, in the context of general models of the effect of grazing on nutrient cycling and for the study system.

I find the redaction a little bit entangled. There are several paragraphs along the manuscript that are not clearly written, wording should be in general revised (see suggestions in Specific comments). I recommend that the manuscript be reviewed by a native English-speaking. I think that the authors could improve the manuscript with several structural and linguistic changes which require a substantial amount of work.

Specific comments

Title

I suggest replace "elevates" by "increase", and "slows" by "decrease"

Abstract

The wording of the abstract should be revised in order to simplify the reading of it.
L. 12. For example, the authors could writing as follow: Litter decomposition and N release are key processes that determine strongly nutrient cycling, but still lack a clear understanding of how grazing affect these processes in alpine ecosystems.
L. 15. I suggest change by: In grazing (GP) and grazing exclusion paddocks (GEP) we identified litter species composition (palatable and unpalatable), and we measure litter quality and soil chemical characteristics. We also measure litter decomposition, using the litter bags methods, and N release in the paddocks over 799 days.
L. 18. Results are a little bit entangled. I suggest describe the pattern found in one of the treatment and compare with the other. For example, In grazed paddock the biomass of palatable plant species was lower than in ungrazing paddock, however the biomass of unpalatable plants was similar. The N and C content of the litter collected in grazing paddock was higher than in the litter of ungrazed paddock.
L. 20-21. Please, review the wording of this sentence.
L. 22-23. Please, remove this last sentence. I suggest conclude about the results found, and highlight the novelty of own work.

Introduction

The wording of this section should be revised.
L. 28. What kind of ecosystems?, If the authors refer to grasslands systems, I suggest directly write "grasslands systems".
L.30. What is the meaning of "degradation rates"?, Soil degradation?, Soil erosion rates??, please, clarify.
L. 34. What are the authors referring to "soil property"?, Fertility?, Organic matter content?, Nutrients availability?, please, clarify.
L. 34. Please, add that grazing have an important impact on the structure and functioning of the ecosystem, because the changes in vegetation communities and soil structure and processes, which affect nutrient cycling.

L. 34-36. Please, add (as I suggest in the Abstract) "…but still lack a clear understanding of how grazing affect these processes in alpine ecosystems."

L. 40. Please change "overall litter quality" by "plant tissues, which is translate to litter quality".

L. 43. Change "loss" by "consume"; and delete "caused" in L. 44

L. 47. Change "will" by "may" (Please, check thoroughly the verb tense used throughout the manuscript)

L. 49. Change "concentrate on" by "consume"

L. 50. Change "will favor" by "promote the", and change "by the" by "of"

L. 53. Change "soil nutrient cycle" by "decomposition"

I suggest delete "still scarce", because there are a lot of works about the relationships between litter quality and decomposition.

In the two follow paragraph the authors describe how litter quality and environmental conditions affect litter decomposition, but do not describe how grazing affect these controls. Please, add.

L. 59. Add "litter" between N and content

L. 60. Add "litter" between lignin and content

L. 63. Change "Except" by "Additionally"

L. 66-73. I suggest that the authors re-write this paragraph, the ideas are mixed. Climate regulates decomposition process at global and regional scale, but microclimate (e.g., soil temperature and moisture) regulates decomposition process at local scale, through influence on microbial activity. At this scale, microclimate and litter quality interact strongly and the rates of decomposition are difficult to predict.

L. 74. I suggest write as follow: "Most of the studies that evaluate the effect of grazing on litter decomposition usually are focused on……" (References).

L. 75-77. I suggest delete this sentence.

L. 79. I suggest write as follow: "Then, we investigate how litter quality affect litter decomposition….." Please, delete "by collecting litter mixtures….", it is not necessary here.

L. 81. I suggest write as follow: We testes the following hypothesis (1) Grazing improved litter quality (i.e., litter with higher nutrient content as N) and promote plant communities with lower biomass of palatable plant species and higher biomass of unpalatable plant species, (2) Grazing increase litter decomposition and N release and thus improve soil properties.
The follow sentence can be removed.

Reading the hypothesis exposed by the authors, I noted that in the first the idea is not clear. The palatable plants (that have higher litter quality) are consumed by herbivores, and then unpalatable plant dominates de community. If grazing promote the abundance and litter biomass of unpalatable plants, how could improve the litter quality of the community??
That is the reason because the second hypothesis contradicts the first. If grazing promote the dominance of plant species with poor litter quality, how could increase nutrient cycling???
Please, clarify, is really important that the hypotheses are well expressed.

Material and methods

L. 91. Change "typical" by "an"

L. 95. Before the reference, add (Supplemental Fig. 1), and delete the "…and the mean temperature and…."

L. 97. What are the authors referring to "including experimental and buffer areas"?, please, clarify.

L. 99. Please review this sentence: "…soil attributes in the experimental area were similar after long-term …", what is the meaning of it?

L. 101. I suggest add a Table or a description of the main soil characteristic.

L. 112. I suggest write as follow: "We collected all plant litter from each quadrat of the GP and GEP for two purposes:…"

(*) Please, I suggest clarify if the quadrats were previous cleaning of litter (i.e., at the moment that the paddock were established) before the harvest. If not, the litter collected is the accumulated litter and no the annual produced litter. This could be a big mistake.

L 118-120. I suggest remove this sentence.

L. 124-128. Please, remove. The Walkley-Black method is usually used for determination of organic C, it is not necessary it description.

L. 131. I suggest remove this sentence, it is not necessary.

L. 133. I suggest write as follow: "We randomly collected five soil samples in each grazing paddock (n=30 in total) at 0-10 cm depth using a …."

L. 156. Deleted "packed"

L. 165. Litter quality or litter quantity??? Please, check.

L. 168. "so on"…???

Please, remove from this section the references to Tables or Figures.

L. 176-183. It is not clear for me the data used in the regression analysis. I understood that the authors used the data of litter decomposition (GE-GEP and GEP-GP) as a result of the soil environment effect, which denominates "site", but, What data are used as a result of litter quality??? For example in Vaieretti et al. (2013) (which the authors refer to perform this analysis), the decomposability of litter is used as an expression of litter quality

In situ litter decomposition is the dependent variable, as well as N release.

Please, clarify this analysis and data used, is really important.

Results

L. 189. Please, based on the comment referred with (*), check the term "annual litter mass", or change by "accumulated litter biomass".

L. 189-192. Why the authors described the differences between palatable and unpalatable plants performed with ANOVA analysis?, according to Statistical analysis section, these differences were tested using paired t-test. Please, check the entire paragraph.

L. 196. I suggest write as follow: "Litter collected from GP…..(Table 1)…."

L. 197. I suggest write as follow: "No significant differences were found for the rest of litter quality characters measured.

L. 199. All these characters were lower in GP compared with GEP, but the differences were not significant. Please, modify this sentence.

L. 207-212. Why the authors describe the dynamic of litter decomposition?, It is not an objective of this work. The same comment for N release. It has not sense analyses differences among the different incubations periods.

I suggest describe the percentage of mass remaining, for example, in the first year, and then for the second year, or the total mass loss in each treatment, and compare the curves.

I suggest change the columns of Figure 3 and Figure 4 by points (with SE) joined by a line.

It is interesting that the $k$ is higher in GP-GP than GEP-GEP, but the mass remaining at the end of the experiment is similar. Moreover, the rate of litter decomposition ($k$) of GP litter was higher when was incubated in GP than in GEP (Home field advantage?), but the rate of decomposition of GEP litter similar in both paddocks, although the mass remaining of GEP was higher in GEP than GP.

The results are really interesting, please, describe and discuss deeply the patterns found.

L. 219. I suggest remove the first sentence.

L. 227-238. See comment for Lines 176-183. I can't evaluate this result if is not clear the data used on it.

Discussion

L. 245-246. The authors did not perform an analysis to evaluate differences in species composition. Please, modify this sentence.

L. 253-254. I suggest write as follow: The low mass of palatable species could be mainly attribute to, one the one hand, on Q-Tibetan Plateau grazing maintained short plant height and …… On the other hand, …….."

At the same time, in this paragraph there are contradicts ideas. First the authors say that palatable plants are short, but immediately later the author say that, in the Q-Tibetan Plateau the most palatable plants are tall. Please, clarify. I suggest that the authors write explicitly the importance of the results found.

L. 264-271. Please, check the wording of the entire paragraph.

Is true that palatable species showed higher litter quality (mainly in terms of C and N content) in GP, however their biomass was significantly lower in this paddock. I suggest discuss these results and how could influence the soil nutrients availability in grazing paddocks compared to ungrazing paddocks.

L. 274-322. All this section is really confused. The authors discuss the effect of climate, which is not sense here, and also the dynamics of litter decomposition.

I suggest the authors concentrate in the comparison between litter decomposition in grazing and ungrazed paddocks, and its relation with litter quality and soil characteristics. The results are really interesting, but the discussion of the pattern found is really poor.

The same comment for "N release" section. Please, check the wording of the sentences, and discuss the patterns found with focus on the effect of grazing on N release.

L. 345-350. This paragraph describes a relationship between litter decomposition and relative litter N concentration, but is not discuss about the mechanisms.

L. 360-369. What the authors refer with "moderate stocking rate"? The treatments in this work were granzing and ungrazing paddock.

Please, highlight the novelty of the work.

Table titles
I suggest write as follow:
Table 1. Initial chemical characteristic (mean ± SE) of litter collected in grazing paddock (GP) and ungrazed paddock (GEP). Different letter indicate significant differences at P < 0.05 level.
Table 2. Delete "Estimated"
Table 3. Delete "Estimated"

Figures captions
I suggest write as follow:
Fig. 1. Delete "Estimated". Please, see the comment regarding to annual or accumulated litter, and correct accordingly.
Fig. 2. (a) Soil total N, (b) soil total P and (c) soil organic C (SOC) content in the grazing (GP) and ungrazing paddock (GEP) in 2010, 2011 and 2012. Asterisk (*) denotes significant differences between grazing paddocks at level P< 0.05.
Fig. 3 and 4. Change to graphics of point connected by a line.

---

## Referee Comment (RC2) · Anonymous Referee #2 · 2 May 2018

This is a very interesting paper. It is worth for publishing and suitable for this journal. The authors investigated the decomposition of litter mix through incubating'in situ' and across environmental conditions over 800 days, providing significant insight into the general nutrient cycling in the alpine ecosystems. The Introduction has provided sufficient background information for the importance of this work. Experimental design is clear. Data analysis and result presentation are appropriate. The authors have logically discussed and interpreted the main findings.However, I have also made a few specific comments or suggestions to improve this manuscript as outlined below:

Lines 20-21 change 'Incubation site environment had more but litter source had less impact on litter decomposition and N release' to 'Incubation site environment had more impact on litter decomposition and N release than did litter source'.

Lines 26-27, remove 'in China' in line 26 and allocate after 'major natural pastures' in line 27.

Lines39 and 287,replace 'pastures' with 'grasslands'. As pastures are different from rangelands which include grasslands, shrublands, woodlands and/or wetlands that grow primarily native vegetation and are often less managed, whilepastures are more intensively managed through seeding, mowing and fertilization.

Line 81, change to 'We tested the hypotheses'.

Lines 82 and 83, delete 'whether'.

Line 87, replace 'Qinghai-Tibetan Plateau' with 'QTP' and afterward, as the abbreviationis a commonly acceptable.

Line 105, should GP be '100 m × 200 m' and GEP be '30 m × 20 m'? Please check again.

Line 114, change 'litter of different species" to 'litter of different species from each quarter'.

Lines 114-117, you had presented the data of dry weight of palatable and unpalatable species, but it was not clear how you did.

Lines 146-147,did you do the same when measured the dry weight of palatable and unpalatable species? Otherwise it should be mentioned previously.

Lines 151-152, change '...small animals (Vaieretti et al., 2013). The small animals were the plateau pika, Ochotona curzoniae (Hodgson) in the present study'to '...small animals (Vaieretti et al., 2013), such as the plateau pikaOchotona curzoniae (Hodgson) in the present study'.

Line 153, change 'Three litter bags' to 'Three litter bags from each treatment'.

Line 156, 'a total of 144 packed litter-bags' should be 24 bags/treatment x 4 treatments

= 96?

Line 164, delete 'those'.

Line 170, add the unit of decay rate (k), g/day.

Line 172, change 'a is the initial litter mass' to 'a is the initial litter mass (i.e., 10 g in this study)'.

Line 196, as stated in lines 180-120, should read 'litter collected from GP' as 'GP-litter',and read 'litter collected from GEP' as 'GEP-litter' Please revise in the relevant cases afterward.

Line 211, change 'faster' to 'shorter'.

Line 254, change 'will can in' to 'will result in'.

Line 268, change 'some degree' to 'a certain extent'.

Line 287, change 'applied in' to 'applied to'.

Lines292-295, a recently published paper (Liang et al. 2018. Grass Forage Sci.) could be cited.

Line 296, change 'the the' to 'the'.

Line 310, change 'short periodof time' to 'short period'.

Line 321, I couldn't find such data from the Table 2, but assume that the authors had converted the k unit from g/day to g/year.

Line 332, add 'in litter' after 'remaining'.

Line 337, add 'the' after 'increases'.

Lines345-346, change 'the greater' to 'the greater the', change 'the faster' to 'the faster the'.

Figure 5, please present the Treatments in Fig. 5b but not in Fig. 5a.

[Figure]

---

## Editor Comment (EC1) · J.-A. Subke (Editor) · 3 May 2018

Dear Professor Hou and co-authors,

This manuscript is generally well written, but I agree with referee 1 that a native or otherwise proficient English speaker should proofread it as part of your revision. Both referees raise a number of detailed queries, which you should address in detail. I would like to add here some additional points to ensure that the objective and approach of your study are presented as clearly as possible. This concerns mainly how you introduce the aims of your study. These should be much more clearly identified and presented in form of hypotheses. Please see my specific comments below on this, and I am happy to elaborate in case you are unsure about how to implement these

changes.

36-39: Delete sentence starting "We carried out...". The introduction should present the broad background and significance underlying this research. Focus on your experiment only at the end of the introduction when presenting your hypotheses.

77-81: Also here, rather than outlining the detail of the experimental design, focus on the hypotheses you want to test, The subsequent sentences give your aims, but they should be more focused (see comments by referee 1). For example, you refer to "improved soil properties" in line 84. It is not clear what this actually means. Be specific which properties you hypothesise to be affected by grazing. Rather than using words such as "improve", make clear which characteristics you test in your approach, and whether you hypothesise an increase or decrease.

571-585: All of the figure captions should have treatments and parameters explained. So avoid referring to GP, GEP, TN, LSD etc. without explaining it here.

Figures 3 and 4: These results are already presented in form of k-values in Table 2. If you think that presenting these data in graph form is at all valuable, I suggest you reduce this to one panel per figure, with all treatments shown as separate lines in the same graph.

Jens-Arne Subke

―――――――――――――――――――――――

---

## Author Comment (AC1) · 21 May 2018

Response to referee #1

General comments

Referee: This study aims to test the effect of grazing on litter decomposition and N release in an alpine Tibetan grassland system. The focus of the manuscript is interesting, and can contribute to the management practices at local and regional scale. However, the authors did not show the novelty found with this work. I think that the authors need to express explicitly in the manuscript the importance of the results found, in the context of general models of the effect of grazing on nutrient cycling and for the study system.

[Figure]

Our response: We greatly appreciate your thoughtful comments. We have strengthened the importance and highlighted the novelties of our work in the revised Abstract and Introduction before the hypotheses. We have also improved the context of the multivariate regression model that addressed the different effects of incubation site and litter quality on litter decomposition and N release. The importance of results from the model has been fully discussed.

Referee: I find the redaction a little bit entangled. There are several paragraphs along the manuscript that are not clearly written, wording should be in general revised (see suggestions in Specific comments). I recommend that the manuscript be reviewed by a native English-speaking. I think that the authors could improve the manuscript with several structural and linguistic changes which require a substantial amount of work.

Our response: According to your constructive and thoughtful comments and suggestions, we have made major revisions on the Abstract, Introduction, Discussion and Conclusions. The revised manuscript has been polished by native English speaker, Professor Corry Matthew (a pasture agronomist).

Specific comments

Title (1) Referee: I suggest replace "elevates" by "increase", and "slows" by "decrease"

Our response: Thanks. As suggested by you and Prof Matthew, we changed the title to "Grazing increases litter decomposition rate but decreases nitrogen release rate in an alpine meadow". Abstract (2) Referee: The wording of the abstract should be revised in order to simplify the reading of it. L. 12. For example, the authors could writing as follow: Litter decomposition and N release are key processes that determine strongly nutrient cycling, but still lack a clear understanding of how grazing affect these processes in alpine ecosystems.

Our response: Thanks, we agree. We have re-worded this sentence as "Litter decomposition and N release are the key processes that strongly determine the nutrient

cycling at the soil-plant interface; however, how these processes are affected by grazing or grazing exclusion in the alpine grassland ecosystems on the Qinghai-Tibetan Plateau (QTP) is poorly understood." Please see lines 24-26 in the revised MS.

(3) Referee: L. 15. I suggest change by: In grazing (GP) and grazing exclusion paddocks (GEP) we identified litter species composition (palatable and unpalatable), and we measure litter quality and soil chemical characteristics. We also measure litter decomposition, using the litter bags methods, and N release in the paddocks over 799 days.

Our response: Thanks. In this study, litter species composition was relevant to the plant species presented in both GP and GEP, and the mass of palatable and unpalatable species was more relevant to litter quality. To eliminate the imprecision we improved the sentences (lines 15-17 in the submitted MS) as "This work studied the short-term (6 month) effects of grazing and grazing exclusion on plant community composition (i.e., plant species presented) and litter quality, and long-term (27 - 33 month) effects on soil chemical characteristics and mixed litter decomposition and N release on the QTP.". Please see lines 29-32 in the revised MS.

(4) Referee: L. 18. Results are a little bit entangled. I suggest describe the pattern found in one of the treatment and compare with the other. For example, In grazed paddock the biomass of palatable plant species was lower than in ungrazing paddock, however the biomass of unpalatable plants was similar. The N and C content of the litter collected in grazing paddock was higher than in the litter of ungrazed paddock.

Our response: We appreciate the referee's suggestion. To make these sentences clear, we improved them as "Our results demonstrate that: (1) shorter term grazing exclusion had no effect on plant community composition but increased plant palatability and total litter biomass; (2) grazing resulted in higher N and C content in litter; and (3) grazing accelerated litter decomposition, while grazing exclusion promoted N release from litter and increased soil organic carbon. In addition, incubation site had significantly more
impact than did litter quality on litter decomposition and N release, while litter quality affected decomposition in the early stages." Please see lines 32-37 in the revised MS.

(5) Referee: L. 20-21. Please, review the wording of this sentence.

Our response: Thanks. We have re-worded this sentence when revised the Abstract. Please see lines 35-37 in the revised MS.

(6) Referee: L. 22-23. Please, remove this last sentence. I suggest conclude about the results found, and highlight the novelty of own work.

Our response: Thanks. By following the referee's comment, we have highlighted the novelty and importance of this study (lines 24-29 in the revised MS) and summarised the main findings in the Abstract (lines 32-37 in the revised MS).

Introduction (7) Referee: The wording of this section should be revised. L. 28. What kind of ecosystems?, If the authors refer to grasslands systems, I suggest directly write "grasslands systems".

Our response: Thanks. We revised as suggested. Please see line 49 in the revised MS.

(8) Referee: L.30. What is the meaning of "degradation rates"?, Soil degradation?, Soil erosion rates??, please, clarify

Our response: Thanks. Now we changed "with an increasing degradation rate of 1.2-7.44 % per year" to "and with the degraded land area increasing at 1.2-7.44 % per year". Please see lines 51-52 in the revised MS.

(9) Referee: L. 34. What are the authors referring to "soil property"?, Fertility?, Organic matter content?, Nutrients availability?, please, clarify.

Our response: Thank you for your comment. We changed "soil property" to "soil organic matter content and nutrient availability" in the MS. Please see line 55 in the revised MS.

(10) Referee: L. 34. Please, add that grazing have an important impact on the structure and functioning of the ecosystem, because the changes in vegetation communities and soil structure and processes, which affect nutrient cycling.

Our response: We appreciate the referee's suggestion which improves the logical link between the sentences. We added the following sentence: "It is well known that grazing may change the vegetation community structure, soil structure and nutrient cycling processes, and that such changes have important consequential impacts on the structure and functioning of the ecosystem as a whole." We also subsequently changed "in the soil-plant interface" in line 35 in the submitted MS to "at the soil-plant interface through grazing". Please see lines 56-60 in the revised MS.

(11) Referee: L. 34-36. Please, add (as I suggest in the Abstract) "...but still lack a clear understanding of how grazing affect these processes in alpine ecosystems."

Our response: Thanks. We revised this sentence as "However, litter decomposition and N release, the key factors regulating the nutrient cycle and availability at the soil-plant interface through grazing (Carrera and Bertiller, 2013), are as yet litter studies in these alpine ecosystems (Luo et al., 2010; Zhu et al., 2016)." Please see lines 58-61 in the revised MS.

(12) Referee: L. 40. Please change "overall litter quality" by "plant tissues, which is translate to litter quality".

Our response: Thanks, we have improved this sentence as "Herbivore grazing may induce short-term ecophysiological changes in plant tissues which in turn may translate into litter quality changes, and long-term shifts in plant community composition." Please see lines 62-63 in the revised MS.

(13) Referee: L. 43. Change "loss" by "consume"; and delete "caused" in L. 44

Our response: Thank you very much. We have revised the sentence as "because the consuming of plant tissues by herbivores may favour…..". Please see lines 65-66 in

the revised MS.

(14) Referee: L. 47. Change "will" by "may" (Please, check thoroughly the verb tense used throughout the manuscript)

Our response: Thank you very much. We agree and made necessary changes throughout the MS accordingly.

(15) Referee: L. 49. Change "concentrate on" by "consume"

Our response: Thanks. We changed "concentrate on" to "preferentially feed on". Please see line 71 in the revised MS.

(16) Referee: L. 50. Change "will favor" by "promote the", and change "by the" by "of"

Our response: Thank you very much. We changed "will favour dominance by the less unpalatable species" to "may promote dominance of unpalatable species". Please see line 72 in the revised MS.

(17) Referee: L. 53. Change "soil nutrient cycle" by "decomposition" I suggest delete "still scarce", because there are a lot of works about the relationships between litter quality and decomposition.

Our response: We appreciate your comment. We agree and revised the sentence as "Empirical evidence of variance in litter quality input and decomposition caused by grazing is still subject to debate (Garibaldi et al., 2007)." Please see lines 78-79 in the revised MS.

(18) Referee: In the two follow paragraph the authors describe how litter quality and environmental conditions affect litter decomposition, but do not describe how grazing affect these controls. Please, add.

Our response: The referee's comment is correct. These two paragraphs described how litter quality and environmental conditions affect litter decomposition. The previous paragraph (lines 44-59 in the revised MS) had provided relevant information about

the effects of grazing on litter decomposition. To eliminate the referee's concern, we added two sentences in that paragraph (lines 74-78 in the revised MS): Therefore, litter in grassland subject to long-term grazing may decompose more slowly. However, some studies demonstrate that grazing per se may accelerate litter decomposition by modifying site conditions for litter turnover through physical changes in the soil by herbivore activities, such as trampling and urine/dung deposition (Takar et al., 1990; Fahnestock and Knapp, 1994; Semmartin et al., 2008; Luo et al., 2010; Liang et al., 2018)..

(19) Referee: L. 59. Add "litter" between N and content

Our response: Thanks. We changed "N content" to "litter N content". Please see line 84 in the revised MS.

(20) Referee: L. 60. Add "litter" between lignin and content

Our response: Thanks. We changed "lignin content" to "litter lignin content". Please see line 85 in the revised MS.

(21) Referee: L. 63. Change "Except" by "Additionally"

Our response: Thanks. "Except" is not correct, we changed "Except" to "In addition to". Please see line 88 in the revised MS.

(22) Referee: L. 66-73. I suggest that the authors re-write this paragraph, the ideas are mixed. Climate regulates decomposition process at global and regional scale, but microclimate (e.g., soil temperature and moisture) regulates decomposition process at local scale, through influence on microbial activity. At this scale, microclimate and litter quality interact strongly and the rates of decomposition are difficult to predict.

Our response: We appreciate the referee's constructive comment. We re-wrote this paragraph accordingly. Please see lines 91-99 in the revised MS.

(23) Referee: L. 74. I suggest write as follow: "Most of the studies that evaluate the effect of grazing on litter decomposition usually are focused on. . . . . ." (References).

Our response: Thanks. We revised this sentence as "Most studies evaluating the effect of grazing on litter decomposition usually focus on forest, grassland or crop ecosystems in temperate areas (e.g., Aber and Melillo, 1980; Berg and Staaf, 1981; Luo et al., 2010; McCurdy et al., 2013), largely ignoring those in the alpine zones.". Please see lines 100-102 in the revised MS.

(24) Referee: L. 75-77. I suggest delete this sentence.

Our response: We agree and deleted this sentence in the revised MS.

(25) Referee: L. 79. I suggest write as follow: "Then, we investigate how litter quality affect litter decomposition….." Please, delete "by collecting litter mixtures….", it is not necessary here.

Our response: We agree. Now we descripted experimental design in one sentence "In this study, we examined the short-term effect (6 month) of grazing and grazing exclusion on plant community composition and litter quality and their longer-term effect on mixed litter decomposition and N release (27 month) and soil chemical characteristics (33 month)". Please see lines 109-111 in the revised MS.

(26) Referee: L. 81. I suggest write as follow: We testes the following hypothesis (1) Grazing improved litter quality (i.e., litter with higher nutrient content as N) and promote plant communities with lower biomass of palatable plant species and higher biomass of unpalatable plant species, (2) Grazing increase litter decomposition and N release and thus improve soil properties. The follow sentence can be removed.

Our response: We agree and revised this part as "Based on the above, this research aimed to test three hypotheses: (1) short-term grazing exclusion does not change plant community composition and litter quality (i.e., nutrient content as N and biomass of palatable plant species), (2) grazing may accelerate litter decomposition and N release and thus increase soil organic carbon and N, and (3) litter quality has less effect on litter decomposition and N release compared to incubation site." Please see lines 111-115

in the revised MS.

(27) Referee: Reading the hypothesis exposed by the authors, I noted that in the first the idea is not clear. The palatable plants (that have higher litter quality) are consumed by herbivores, and then unpalatable plant dominates de community. If grazing promote the abundance and litter biomass of unpalatable plants, how could improve the litter quality of the community?? That is the reason because the second hypothesis contradicts the first. If grazing promote the dominance of plant species with poor litter quality, how could increase nutrient cycling??? Please, clarify, is really important that the hypotheses are well expressed.

Our response: The referee's comments are right. We have improved the hypotheses accordingly. Please see our last response (26) or lines 111-115 in the revised MS.

Material and methods (28) Referee: L. 91. Change "typical" by "an"

Our response: We have changed. Please see line 121 in the revised MS.

(29) Referee: L. 95. Before the reference, add (Supplemental Fig. 1), and delete the "...and the mean temperature and...."

Our response: Thanks. We revised as "(Sun et al., 2015; Supplemental Fig. 1). Please see line 125 in the revised MS.

(30) Referee: L. 97. What are the authors referring to "including experimental and buffer areas"?, please, clarify.

Our response: Thinks. We have clarified as "The grassland selected for experiments was > 9 ha in area (including 6 ha of experimental plots and 3 ha buffer areas)". Please see lines 126-127 in the revised MS.

(31) Referee: L. 99. Please review this sentence: "...soil attributes in the experimental area were similar after long-term ...", what is the meaning of it?

Our response: Thinks. We changed "soil attributes" to "soil properties". Please see

lines 128-129 in the revised MS.

(32) Referee: L. 101. I suggest add a Table or a description of the main soil characteristic.

Our response: The details of soil characteristics are available in the cited paper (Wu et al. 2010).

(33) Referee: L. 112. I suggest write as follow: "We collected all plant litter from each quadrat of the GP and GEP for two purposes:..."

Our response: According to the referee's comment, we have re-written this sentence as: In October 2010, we collected all plant litter from each quadrat of the GP and GEP for two purposes: (1) measurement of litter composition and quality in this experiment, and (2) measurement of litter decomposition and N release in the next experiment. Please see lines 138-141 in the revised MS.

(34) Referee: (*) Please, I suggest clarify if the quadrats were previous cleaning of litter (i.e., at the moment that the paddock were established) before the harvest. If not, the litter collected is the accumulated litter and no the annual produced litter. This could be a big mistake.

Our response: Yes. Litter was cleared at the moment when the paddocks were established. Now we have clarified. Please see lines 137-138 in the revised MS.

(35) Referee: L 118-120. I suggest remove this sentence.

Our response: Thanks. We removed it and clarified the experimental design. Lines 145-149 the revised MS.

(36) Referee: L. 124-128. Please, remove. The Walkley-Black method is usually used for determination of organic C, it is not necessary it description.

Our response: We agree and removed it.

(37) Referee: L. 131. I suggest remove this sentence, it is not necessary.

Our response: We agree and removed it.

(38) Referee: L. 133. I suggest write as follow: "We randomly collected five soil samples in each grazing paddock (n=30 in total) at 0-10 cm depth using a ...."

Our response: Thanks. We revised as "We randomly collected five soil samples in each experimental paddock (n=30 in total) from the 0 - 10 cm depth using a bucket auger...". Please see lines 158-160 in the revised MS.

(39) Referee: L. 156. Deleted "packed"

Our response: Thanks. We deleted it as well as that in line 152 in the submitted MS.

(40) Referee: L. 165. Litter quality or litter quantity??? Please, check.

Our response: It was litter quality. Now we have changed "Data on litter quality" to "Data on the initial chemical characteristics of litter". Please see line 190 in the revised MS.

(41) Referee: L. 168. "so on"...???

Our response: We changed "so on" to "other chemical characteristics of litter". Please see line 195 in the revised MS.

(42) Referee: Please, remove from this section the references to Tables or Figures.

Our response: We suggest to remain those references so as the readers could easily to track the methods used to analyses the data.

(43) Referee: L. 176-183. It is not clear for me the data used in the regression analysis. I understood that the authors used the data of litter decomposition (GE-GEP and GEP-GP) as a result of the soil environment effect, which denominates "site", but, What data are used as a result of litter quality??? For example in Vaieretti et al. (2013) (which the authors refer to perform this analysis), the decomposability of litter is used as an

expression of litter quality In situ litter decomposition is the dependent variable, as well as N release. Please, clarify this analysis and data used, is really important

Our response: Thank you very much for these comments. We have improved as "A multivariate regression model (GLM Procedure) employed by Vaieretti et al. (2013) was applied to quantify the effect of incubation site and litter quality (the two independent factors) on the final litter decomposition or N release (the dependent factor) (Table 3): litter decomposition or N release = Site + Quality + Site$\times$Quality + Ïţ, where 'Site' is the paddock category where the litter was incubated (i.e., incubation site: GP and GEP), 'Quality' is the litter quality reflecting the sources where the litter was collected from (i.e., GP and GEP), and Ïţ is the model error." Hope it is clear now (also see lines 204-209 in the revised MS).

Results (44) Referee: L. 189. Please, based on the comment referred with (*), check the term "annual litter mass", or change by "accumulated litter biomass".

Our response: Thanks. The original "annual litter mass" was correct. Please see line 219 in the revised MS.

(45) Referee: L. 189-192. Why the authors described the differences between palatable and unpalatable plants performed with ANOVA analysis?, according to Statistical analysis section, these differences were tested using paired t-test. Please, check the entire paragraph.

Our response: We appreciate the referee's careful checking. The method applied to compare the difference in biomass of palatable species, biomass of unpalatable species and biomass of total biomass between GP and GEP was missing. Now we improved as "Data on the biomass of palatable or unpalatable species and that on the total biomass between GP and GEP were also analyzed using ANOVA, while for GP or GEP the difference in litter biomass between the palatable and unpalatable species was compared by paired-t test (TTEST Procedure) (Fig. 1)". Please see lines 191-194 in the revised MS. The results reported in the submitted MS were correct.

(46) Referee: L. 196. I suggest write as follow: "Litter collected from GP.....(Table 1)...."

Our response: We agree. We deleted "As shown in Table 1" and revised the sentence as "GP-litter had significantly higher C and N but significantly lower hemicellulose and hemicellulose:N than GEP-litter (Table 1)". Please see lines 226-227 in the revised MS.

(47) Referee: L. 197. I suggest write as follow: "No significant differences were found for the rest of litter quality characters measured.

Our response: Thanks. We merged this and next comments and revised as "Although other quality characteristics were lower in GP-litter than in GEP-litter, the differences were not significant (Table 1).". Please see lines 227-228 in the revised MS.

(48) Referee: L. 199. All these characters were lower in GP compared with GEP, but the differences were not significant. Please, modify this sentence.

Our response: Same as Comment (47). We have revised. Please see lines 227-228 in the revised MS.

(49) Referee: L. 207-212. Why the authors describe the dynamic of litter decomposition?, It is not an objective of this work. The same comment for N release. It has not sense analyses differences among the different incubations periods. I suggest describe the percentage of mass remaining, for example, in the first year, and then for the second year, or the total mass loss in each treatment, and compare the curves. I suggest change the columns of Figure 3 and Figure 4 by points (with SE) joined by a line. It is interesting that the k is higher in GP-GP than GEP-GEP, but the mass remaining at the end of the experiment is similar. Moreover, the rate of litter decomposition (k) of GP litter was higher when was incubated in GP than in GEP (Home field advantage?), but the rate of decomposition of GEP litter similar in both paddocks, although the mass remaining of GEP was higher in GEP than GP. The results are really interesting, please, describe and discuss deeply the patterns found.

Our response: We appreciate the thorough comments made by the referee. We have removed the detailed descripts of the dynamic of litter decomposition. As suggested, we did not analyse the data of litter decomposition as well as that of N release amount the different incubation periods. We compared the difference in litter decomposition between the first and second years (lines 236-239) and discussed the mechanisms (lines 319-332 in revised MS); however, as N release fluctuated during the incubation period, the difference between the first and second years was not compared. According to Dr. Subke's and the referee's suggestions, we have replaced the columns with a joined-point line for each treatment and placed those lines in one panel for litter decomposition or N release. We merged Figures 3 and 4 in the submitted manuscript into one figure (Fig. 3) in the revised ones. As k values were estimated by all data collected during the incubation period, it may be not completely linked the k to the data collect at a specific time point.

(50) Referee: L. 219. I suggest remove the first sentence.

Our response: We removed the first sentence as requested.

(51) Referee: L. 227-238. See comment for Lines 176-183. I can't evaluate this result if is not clear the data used on it.

Our response: We have improved the model descriptions by considering the referee's comment (43) made for Lines 176-183 in the submitted MS. We have also made corresponding revisions in Section 3.4 (please see lines 253-264 in the revised MS). We hope it is clear now.

Discussion (52) Referee: L. 245-246. The authors did not perform an analysis to evaluate differences in species composition. Please, modify this sentence.

Our response: As we had shown in the Results (lines 222-224 in the revised MS) that of 55 species identified in GP, only one species was not found in GEP. Such data must be not significantly different and thus are not subjected to analysis, otherwise a chi-square

test can be used.

(53) Referee: L. 253-254. I suggest write as follow: The low mass of palatable species could be mainly attribute to, one the one hand, on Q-Tibetan Plateau grazing maintained short plant height and ......... On the other hand, ..........." At the same time, in this paragraph there are contradicts ideas. First the authors say that palatable plants are short, but immediately later the author say that, in the Q-Tibetan Plateau the most palatable plants are tall. Please, clarify. I suggest that the authors write explicitly the importance of the results found.

Our response: We appreciate the cogitative comment made by the referee. We have checked again the cited reference and improved the discussion by keeping the second point only. We have also made revisions. Please see lines 279-284 in the revised MS.

(54) Referee: L. 264-271. Please, check the wording of the entire paragraph. Is true that palatable species showed higher litter quality (mainly in terms of C and N content) in GP, however their biomass was significantly lower in this paddock. I suggest discuss these results and how could influence the soil nutrients availability in grazing paddocks compared to ungrazing paddocks.

Our response: Litter used for quality test was the mixed litter. We tested the quality of mixed litter rather than the litter quality of palatable and unpalatable species. We have tried to clarify the experimental design. Please see lines 145-152 in the revised MS.

(55) Referee: L. 274-322. All this section is really confused. The authors discuss the effect of climate, which is not sense here, and also the dynamics of litter decomposition. I suggest the authors concentrate in the comparison between litter decomposition in grazing and ungrazed paddocks, and its relation with litter quality and soil characteristics. The results are really interesting, but the discussion of the pattern found is really poor. The same comment for "N release" section. Please, check the wording of the sentences, and discuss the patterns found with focus on the effect of grazing on N release.

Our response: Thanks. By considering this and previous comments, we have made major revisions in this section by focusing on litter decomposition or N release differs between GP and GEP.

(56) Referee: L. 345-350. This paragraph describes a relationship between litter decomposition and relative litter N concentration, but is not discuss about the mechanisms.

Our response: Thanks. This paragraph becomes redundant after we have made major revisions in this section, thus we removed it.

(57) Referee: L. 360-369. What the authors refer with "moderate stocking rate"? The treatments in this work were granzing and ungrazing paddock. Please, highlight the novelty of the work.

Our response: Thanks. We agree. We have completely re-worded the Conclusion to highlight the novelty of our work. Please see lines 370-378 in the revised MS.

Table titles (58) Referee: I suggest write as follow: Table 1. Initial chemical characteristic (mean ± SE) of litter collected in grazing paddock (GP) and ungrazed paddock (GEP). Different letter indicate significant differences at P < 0.05 level.

Our response: Thanks. The title is revised as: Table 1. Initial chemical characteristics (mean ± SE) of litter collected grazing paddocks (GP-litter) and grazing exclusion paddocks (GEP-litter). Unit of chemical characteristics is mg/g litter for C, N, P, lignin, cellulose and hemicellulose. Different letters in each row indicate significant difference at P < 0.05 level.

(59) Referee: Table 2. Delete "Estimated"

Our response: Yes. We deleted "Estimated" and revised the title as: Table 2. Litter decay rate (k, g·10 g-·day-) in different incubation environments. k values followed by different letters are significantly different (non-overlap of 83.4 % CL). The R2, F and P are estimated from the negative exponential model of Swift et al. (1979).

(60) Referee: Table 3. Delete "Estimated"

Our response: Yes. We deleted "Estimated" and revised the title according to Dr. Subke's comments: Table 3. Contribution (%) of incubation site (Site: GP, grazing paddocks; GEP, grazing exclusion paddocks) and litter quality (Quality: GP-litter, mixed litter collected from grazing paddocks; GEP-litter, mixed litter collected form grazing exclusion paddocks) to litter decomposition and N release..

Figures captions (61) Referee: I suggest write as follow: Fig. 1. Delete "Estimated". Please, see the comment regarding to annual or accumulated litter, and correct accordingly.

Our response: Thanks. Now it is changed to: Figure 1. Mean ($\pm$ SE) annual biomass of litter collected from grazing paddocks (GP) and grazing exclusion paddocks (GEP). For each category, columns with different letters are significantly different (ANOVA: P < 0.05).

(62) Referee: Fig. 2. (a) Soil total N, (b) soil total P and (c) soil organic C (SOC) content in the grazing (GP) and ungrazing paddock (GEP) in 2010, 2011 and 2012. Asterisk (*) denotes significant differences between grazing paddocks at level P< 0.05.

Our response: Thanks. We changed the caption to: Figure 2. Comparison of (a) soil total nitrogen (TN), (b) soil total phosphorous (TP), and (c) soil organic carbon (SOC) between the grazing paddocks (GP) and grazing exclusion paddocks (GEP). *Significant difference was only found between GP and GEP for SOC in 2012 (P < 0.05).

(63) Referee: Fig. 3 and 4. Change to graphics of point connected by a line.

Our response: We have changed according to Dr. Subke's and your comments: Figure 3. Dynamics (mean $\pm$ SE) of litter decomposition (a) and N lease (b) on the QTP. GP-GP, mixed litter was collected from grazing paddocks (GP) and incubated in GP; GEP-GEP, mixed litter was collected from grazing exclusion paddocks (GEP) and incubated
in GEP; GEP-GP, mixed litter was collected from GEP and incubated in GP; GP-GEP, mixed litter was collected from GP and incubated in GEP. Grey lines under months indicate the mean air temperatures < 0 °C.

We hope our responses and revisions made in the revised MS are appropriate.

[Figure]

**Fig. 1.** Figure 1. Mean (± SE) annual biomass of litter collected from grazing paddocks (GP) and grazing exclusion paddocks (GEP). For each category, columns with different letters are significantly different

[Figure]

[Figure]

**Fig. 2.** Figure 2. Comparison of (a) soil total nitrogen (TN), (b) soil total phosphorous (TP), and (c) soil organic carbon (SOC) between the grazing paddocks (GP) and grazing exclusion paddocks (GEP). *Signif

[Figure]

[Figure]

Incubation period (days)

**(a)**

**(b)**

**Fig. 3.** Figure 3. Dynamics (mean ± SE) of litter decomposition (a) and N lease (b) on the QTP. GP-GP, mixed litter was collected from grazing paddocks (GP) and incubated in GP; GEP-GEP, mixed litter was colle

---

## Author Comment (AC2) · 21 May 2018

Response to referee #2

General comments Referee: This is a very interesting paper that I have read in the past years. It is worth for publishing and suitable for this journal. The authors investigated the decomposition of litter mix through incubating 'in situ' and across environmental conditions over 800 days, providing significant insight into the general nutrient cycling in the alpine ecosystems. The Introduction has provided sufficient background information for the importance of this work. Experimental design is clear. Data analysis and result presentation are appropriate. The authors have logically discussed and interpreted the main findings. However, I have also made a few specific comments or suggestions to improve this manuscript as outlined below:

Our response: We appreciate all of the referee's positive comments and thoughtful suggestions. We have accepted all of the suggestions after considering that of Dr. Subke and referee #1. We explain how we have revised the manuscript by following the referee's comments one by one.

Specific comments (1) Referee: Lines 20-21 change 'Incubation site environment had more but litter source had less impact on litter decomposition and N release' to 'Incubation site environment had more impact on litter decomposition and N release than did litter source'.

Our response: Thanks. The main findings of this study have been reworded according to the comments of both referees.

(2) Referee: Lines 26-27, remove 'in China' in line 26 and allocate after 'major natural pastures' in line 27.

Our response: Thanks. We revised accordingly.

(3) Referee: Lines 39 and 287, replace 'pastures' with 'grasslands'. As pastures are different from rangelands which include grasslands, shrublands, woodlands and/or wetlands that grow primarily native vegetation and are often less managed, while pastures are more intensively managed through seeding, mowing and fertilization.

Our response: We thank the thoughtful comments and made corresponding revisions.

(4) Referee: Line 81, change to 'We tested the hypotheses'.

Our response: Thanks. The hypotheses have been improved according to the comments of Dr. Subke and referee #1.

(5) Referee: Lines 82 and 83, delete 'whether'.

Our response: Thanks. The hypotheses have been reworded according to the comments of Dr. Subke and referee #1.

(6) Referee: Line 87, replace 'Qinghai-Tibetan Plateau' with 'QTP' and afterward, as the abbreviation is a commonly acceptable.

Our response: We agree and revised through the manuscript.

(7) Referee: Line 105, should GP be '100 m × 200 m' and GEP be '30 m × 20 m'? Please check again.

Our response: You are right. Thanks. We have revised.

(8) Referee: Line 114, change 'litter of different species" to 'litter of different species collected from each quarter'.

Our response: Thanks. We have improved. Please see line 145 in the revised MS.

(9) Referee: Lines 114-117, you had presented the data of dry weight of palatable and unpalatable species, but it was not clear how you did.

Our response: Thanks. We added one sentence to clarify how to collect these data. Please see lines 148-149 in the revised MS.

(10) Referee: Lines 146-147, did you do the same when measured the dry weight of palatable and unpalatable species? Otherwise it should be mentioned previously.

Our response: We agree. After the revision according last comment, it is clear now. Please see lines 181-182 in the revised MS.

(11) Referee: Lines 151-152, change '...small animals (Vaieretti et al., 2013). The small animals were the plateau pika, Ochotona curzoniae (Hodgson) in the present study'to '...small animals (Vaieretti et al., 2013), such as the plateau pika Ochotona curzoniae (Hodgson) in the present study'.

Our response: Thanks. We revised as suggested. Please see lines 181-182 in the revised MS.

(12) Referee: Line 153, change 'Three litter bags' to 'Three litter bags from each treatment'.

Our response: Thanks. We merged these two sentences and changed to "On 20th Oct 2010, the packed litter was incubated above the soil surface by fastening to the ground surface with four steel stakes to prevent removal by sheep and small animals (Vaieretti et al., 2013), such as the plateau pika Ochotona curzoniae (Hodgson)." Please see lines 147-177 in the revised MS.

(13) Referee: Line 156, 'a total of 144 packed litter-bags' should be 24 bags/treatment x 4 treatments = 96?

Our response: Thank you very much for the carefully checking. We have corrected it. Please see line 180 in the revised MS.

(14) Referee: Line 164, delete 'those'.

Our response: Yes. We deleted it.

(15) Referee: Line 170, add the unit of decay rate (k), g/day.

Our response: Thanks. We have checked again carefully and revised as the journal requires: gÂů10 g-Âůday-. Please see line 197 in the revised MS.

(16) Referee: Line 172, change 'a is the initial litter mass' to 'a is the initial litter mass (i.e., 10 g in this study)'.

Our response: Thanks. We revised accordingly. Please see line 199 in the revised MS.

(17) Referee: Line 196, as stated in lines 180-120, should read 'litter collected from GP' as 'GP-litter', and read 'litter collected from GEP' as 'GEP-litter' Please revise in the relevant cases afterward.

Our response: We appreciate this comment. Should be 'lines 118-120' in the submitted MS. We have made relevant revisions through the manuscript. Please see lines 226- in the revised MS.

(18) Referee: Line 211, change 'faster' to 'shorter'.

Our response: Thanks. We removed this sentence as it was not necessary to present such data.

(19) Referee: Line 254, change 'will can in' to 'will result in'.

Our response: Thanks. We have removed the sentence after considering the referee #1's comment.

(20) Referee: Line 268, change 'some degree' to 'a certain extent'.

Our response: Thanks. We changed to "some extent". Please see line 293 in the revised MS.

(21) Referee: Line 287, change 'applied in' to 'applied to'.

Our response: Thanks. We have removed this sentence in the revised MS.

(22) Referee: Lines 292-295, a recently published paper (Liang et al. 2018. Grass Forage Sci.) could be cited.

Our response: Thanks. We agree and have added this paper. Please see lines 78, 108, 319 and 466 in the revised MS.

(23) Referee: Line 296, change 'the the' to 'the'.

Our response: Thanks. We revised accordingly.

(24) Referee: Line 310, change 'short period of time' to 'short period'.

Our response: We have removed this sentence after the major revisions are made in this section.

(25) Referee: Line 321, I couldn't find such data from the Table 2, but assume that the authors had converted the k unit from g/day to g/year.

Our response: We appreciate this comment. We have revised. Please see lines 313-314 in the revised MS.

(26) Referee: Line 332, add 'in litter' after 'remaining'.

Our response: Thanks. We have made major revisions in the revised MS.

(27) Referee: Line 337, add 'the' after 'increases'.

Our response: Thanks. We have made major revisions in the revised MS.

(28) Referee: Lines 345-346, change 'the greater' to 'the greater the', change 'the faster' to 'the faster the'.

Our response: We have removed this sentence after the major revisions are made in this section.

(29) Referee: Figure 5, please present the Treatments in Fig. 5b but not in Fig. 5a.

Our response: Thank you very much. We have revised accordingly.

We hope our responses and revisions made in the revised MS are appropriate.

Please also note the supplement to this comment:
https://www.biogeosciences-discuss.net/bg-2018-66/bg-2018-66-AC2-supplement.pdf

**Supplement:**

[revised manuscript text omitted]

---

## Author Comment (AC3) · 21 May 2018

21 May 2018

Dr. Jens-Arne Subke Editor Biogeosciences jens-arne.subke@stir.ac.uk
Thank you very much for your letter on 03 May 2018 with two anonymous referees' reports and for your providing an opportunity for us to revise the paper. We have considered all your and referees' comments very carefully and made revisions accordingly.

These constructive comments are greatly appreciated and now acknowledged in the Acknowledgment section. We have highlighted (yellow) our revisions in the manuscript for your convenience.

We follow these comments in order and explain what we have done point by point. We response to the two anonymous referees' comments in separated files (i.e., Response to bg-2018-66-RC1 and Response to bg-2018-66-RC2).

Response to Dr. Subke's comments

Dr. Subke: This manuscript is generally well written, but I agree with referee 1 that a native or otherwise proficient English speaker should proofread it as part of your revision. Both referees raise a number of detailed queries, which you should address in detail. I would like to add here some additional points to ensure that the objective and approach of your study are presented as clearly as possible. This concerns mainly how you introduce the aims of your study. These should be much more clearly identified and presented in form of hypotheses. Please see my specific comments below on this, and I am happy to elaborate in case you are unsure about how to implement these.

Our response: We appreciate all of your and referees' constructive and thoughtful comments. We have accepted all suggestions and revised the paper accordingly. We also explain how we have revised the manuscript. The manuscript has been revised and proofread by a native English speaker, Professor Corry Matthew (a pasture agronomist).

Dr. Subke: 36-39: Delete sentence starting "We carried out. . .". The introduction should present the broad background and significance underlying this research. Focus on your experiment only at the end of the introduction when presenting your hypotheses.

Our response: Thanks. We have deleted this sentence. We have also strengthened the importance and highlighted the novelty of our work before presenting the hypotheses.

[Figure]

Dr. Subke: 77-81: Also here, rather than outlining the detail of the experimental design, focus on the hypotheses you want to test, The subsequent sentences give your aims, but they should be more focused (see comments by referee 1). For example, you refer to "improved soil properties" in line 84. It is not clear what this actually means. Be specific which properties you hypothesise to be affected by grazing. Rather than using words such as "improve", make clear which characteristics you test in your approach, and whether you hypothesise an increase or decrease.

Our response: We completely agree with your comments. We only provided the necessary information relevant to the experimental design and gave clear descriptions of hypotheses.

Dr. Subke: 571-585: All of the figure captions should have treatments and parameters explained. So avoid referring to GP, GEP, TN, LSD etc. without explaining it here. Figures 3 and 4: These results are already presented in form of k-values in Table 2. If you think that presenting these data in graph form is at all valuable, I suggest you reduce this to one panel per figure, with all treatments shown as separate lines in the same graph.

Our response: Thanks. We have detailed the treatments and parameters in captions of all figures and tables. We prefer to keep the results of Figures 3 and 4 in this paper, as the referee #1 suggested to compare the litter decomposition and N release between the first and second year. According to your and referee #1's suggestions, we have replaced the columns with a joined-point line for each treatment and placed those lines in one panel for litter decomposition or N release. We then merged Figures 3 and 4 into one figure (Fig. 3a, b).

We hope our responses and revisions made in the revised MS are appropriate. Please let us know if further revisions are required.

Thank you for considering our paper. Sincerely yours,

Fujiang Hou Professor of Pasture Science College of Pastoral Agriculture Science and Technology Lanzhou University Lanzhou City, Gansu Province, P. R. China E-mail: cyhoufj@lzu.edu.cn

Please also note the supplement to this comment:
https://www.biogeosciences-discuss.net/bg-2018-66/bg-2018-66-AC3-supplement.pdf

---

## Author Response (AR2)

Dear Dr. Subke,

We appreciate your letter on 26 June 2018. We have accepted all of your suggestions and made revisions accordingly. Those revisions have been highlighted (Green) in the final manuscript for your convenience.

| Your suggestions | Revisions in the final manuscript |
|---|---|
| Line 42: "1990s" | Line 41 |
| Line 43: "have recently been…" | Line 43 |
| Lines 123-125: Avoid using brackets if possible. Here for example: "Three sampling methods were designed to minimize the sample variance caused by the uneven litter distribution and to ensure the similar composition and quality of litter used for this and the next experiment: (1) half alongside, (2) half along diagonal, and (3) two sub-quarters (0.25 × 0.25 m) along diagonal (Supplemental Fig. 2)." | Lines 123-125 |
| Line 191: "55", rather than "fifty-five". | Line 191 |
| Line 266: Delete "Firstly". | Line 266 |
| Lines 304-307: this sentence is hard to follow. I suggest the following: "Thus, soil properties are unlikely to be significantly changed through grazing or grazing exclusion over relatively short periods, indicating that limited grazing events have a smaller effect on litter decomposition under cool environments on the QTP than in experiments conducted in warmer climates." | Lines 304-307 |

Thank you very much for considering our paper.

Sincerely yours,

Fujiang Hou

Professor of Pasture Science
College of Pastoral Agriculture Science and Technology
Lanzhou University
Lanzhou City, Gansu Province, P. R. China
E-mail: cyhoufj@lzu.edu.cn